# TYPED CHAIN-OF-THOUGHT: A CURRY-HOWARD FRAMEWORK FOR VERIFYING LLM REASONING

## ABSTRACT

While Chain-of-Thought (CoT) prompting enhances the reasoning capabilities of large language models, the faithfulness of the generated rationales remains an open problem for model interpretability. We propose a novel theoretical lens for this problem grounded in the Curry-Howard correspondence, which posits a direct relationship between formal proofs and computer programs. Under this paradigm, a faithful reasoning trace is analogous to a well-typed program, where each intermediate step corresponds to a typed logical inference. We operationalise this analogy, presenting methods to extract and map the informal, natural language steps of CoT into a formal, typed proof structure. Successfully converting a CoT trace into a well-typed proof serves as a strong, verifiable certificate of its computational faithfulness, moving beyond heuristic interpretability towards formal verification. Our work provides a principled bridge between the emergent, often opaque reasoning of LLMs and the rigorous semantics of formal systems, proposing a new direction for the mechanistic interpretability of complex, multi-step reasoning.

## 1 INTRODUCTION

The interpretability of large language model (LLM) outputs, particularly their faithfulness to verifiable underlying computational processes, represents a fundamental challenge in modern AI research. This challenge has become more acute with the rise of language reasoning models (LRMs), characterised by Chain-of-Thought (CoT) prompting (Wei et al., 2022; Wang et al., 2022) and multi-step reasoning as a means of consistently improving model performance and expressivity Merrill & Sabharwal (2023) across diverse reasoning tasks. The extent to which CoT reflects underlying processes characteristic of computation DeepMind (2025) with potential for monitoring or control Korbak et al. (2025) remains an open question. Other research has problematised the claim that CoT rationales, be they intermediate reasoning traces or final post hoc explanatory artefacts, may not faithfully reflect the model's actual computational process (Turpin et al., 2023; Barez et al., 2025; Sharkey et al., 2025). Such uncertainty raises important questions about whether these explanations serve as reliable windows into model reasoning in ways that could facilitate greater alignment and control of models and ensure the veracity of model outputs, or merely as plausible post-hoc narratives.

Current approaches to this interpretability challenge fall into several categories. Tool-augmented methods utilise external verification architecture for reasoning components (Gao et al., 2022). Structured inference frameworks recast generation as an optimisation search procedure over candidate thoughts (Yao et al., 2023) or impose graph-like structures during decoding (Zhang et al., 2024; Abdaljalil et al., 2025). Formal verification pipelines, common in the growing research on automated and semi-automated proof systems with LRMs, often deploy LLM-based proof assistants and verifiers (Wang et al., 2025; Baba et al., 2025) to translate natural language outputs into formal sequences to check correctness. Yet none of these methods directly type the natural language CoT itself at decode time, nor do they produce per-step typed certificates auditable independently of downstream provers She et al. (2025). Our research question is therefore: when can an interpretation of a language model's reasoning be considered *computationally programmatic*? In particular, can we define and enforce conditions under which CoT traces correspond to well-typed programs whose dataflow provably connects premises to conclusions?

In this work, we explore answers to this question by drawing upon and operationalising the Curry-Howard correspondence (CHC) as a tool for interpretability. The CHC is an isomorphism that holds in certain circumstances between computational programs and mathematical proofs: proofs are programs and propositions are types, underlying modern proof assistants. We argue that in certain cases, reasoning can itself be mapped to a computationally faithful typed program that generates its output, providing both correctness guarantees and a form of computational interpretability.

**Contributions.** To this end, we introduce *Proof-Carrying Chain-of-Thought (PC-CoT)* which provides the following contributions to the literature:

1. *Typed natural language CoT during decoding*, producing per-step Typed Faithfulness Certificates (TFCs) that capture rule applications, type checks, and typed dataflow, in effect a decode-time implementation of the CHC for LLM reasoning.

2. *Constructive Typed Reasoning Graphs (TRGs)* that represent typed dataflow as bipartite graphs between statements and rules. We introduce novel formal metrics (Coverage, Evidence Validity Rate, Path Existence) quantifying typed support for answers.

3. *Certified Self-Consistency (CSC)*, which aggregates only over experiments satisfying typing constraints, achieving 69.8% accuracy on GSM8K versus 19.6% for standard baselines—a 50.3% improvement using identical sampling budgets.

The use of the CHC as an interpretability tool has the benefit that it is in principle applicable at whatever level of abstraction evidence of causal computation is being sought. CHC-based interpretability applies regardless of whether one considers post-hoc CoT, intermediate reasoning steps, or mechanistic circuit representations. Code for our work can be found in the accompanying repository Anonymous (2025).

## 2 RELATED WORK

### 2.1 THE CURRY-HOWARD CORRESPONDENCE

The CHC establishes a fundamental isomorphism between logic and computation: propositions are types, and proofs are programs (Luo, 2011; Pierce et al., 2015) (sometimes called the 'proofs-as-programs' theorem). Formally, a logical implication $P \supset Q$ corresponds to the function type $P \to Q$, where a proof of the implication is a program transforming evidence of $P$ into evidence of $Q$. Under the correspondence we have the following equivalences:

- Conjunction $P \wedge Q$ corresponds to product type $P \times Q$
- Disjunction $P \vee Q$ corresponds to sum type $P + Q$
- Universal quantification $\forall x.P(x)$ corresponds to dependent product $\Pi x.P(x)$
- Existential quantification $\exists x.P(x)$ corresponds to dependent sum $\Sigma x.P(x)$

In a simply typed lambda calculus, this yields a precise equivalence: if $\Gamma \vdash M : A$ in the type system, then there exists a natural deduction proof of $A$ under assumptions $\Gamma$, and conversely, any proof yields a term inhabiting its proposition.

### 2.2 LLMS AND FORMAL REASONING

**The Curry-Howard Correspondence and Proof Assistants.** Modern proof assistants like Coq and Lean operationalise this principle—theorem statements become types, proof scripts construct terms inhabiting those types, and verification reduces to type-checking (Lu et al., 2025; Baba et al., 2025), such as for compilers verifying functional correctness through static type-checking (Wang et al., 2025).

**LLMs for Formal Proof Generation.** Considerable advances in LLM and LRM performance have also seen a significant increase in research seeking to integrate LLMs with formal automated and semi-automated proof systems Trinh & Luong (2024), such as via proof assistants, in order

to produce formal proofs. The PROVER-AGENT framework orchestrates informal reasoning LLMs with Lean feedback, ensuring correctness through type-checking at each inference step (Baba et al., 2025). MA-LoT enhances this approach using long CoT plans in natural language coupled with corrector models informed by Lean feedback, achieving state-of-the-art results on MiniF2F (Wang et al., 2025). While these systems successfully leverage the CHC within proof assistants, they translate natural language chains into formal proofs *post hoc* rather than typing the model CoT reasoning traces during generation.

**Structured Chain-of-Thought Frameworks.** Several frameworks restructure CoT reasoning into more sophisticated search spaces. Tree-of-Thoughts (ToT) explores multiple reasoning paths with self-evaluation and backtracking (Yao et al., 2023). Graph-of-Thoughts (GoT) represents thought units as nodes with edges capturing non-sequential reasoning patterns (Yao et al., 2024). Diagram-of-Thought (DoT) internalizes complex reasoning within single models, constructing directed acyclic graphs grounded in topos theory and interpreting summarization as categorical colimits (Zhang et al., 2024). Theorem-of-Thoughts (ToTh) employs multi-agent frameworks combining abductive, deductive, and inductive reasoning with Bayesian belief propagation (Abdaljalil et al., 2025). While these methods impose valuable structure, they operate purely at the natural language level without CHC typing, relying on heuristic scoring rather than formal type-checking.

**How PC-CoT is unique** PC-CoT adopts a unique approach by applying the CHC as a decoding constraint rather than post-hoc validation method. Each natural language step receives a type via lightweight rule schemas, enabling construction of Typed Reasoning Graphs whose typed dataflow must connect premises to conclusions. Unlike LLM-for-proof pipelines that type subsequent formal scripts, PC-CoT types the natural language itself. Unlike structured CoT frameworks that rely on plausibility heuristics, PC-CoT's certification method grounded in the CHC provides a way to ensure that reasoning traces are accepted only when they can be reinterpreted as well-typed programs.

## 3 METHODS AND NOTATION

### 3.1 LIMITED TYPE SYSTEM FOR CHAIN-OF-THOUGHT

To operationalise the CHC for model reasoning, we introduce a limited type system tailored to arithmetic and logical reasoning (see the Appendix for worked examples and the codebase). Our system includes:

- *Numeric types:* $\mathbb{N} \subseteq \mathbb{Z} \subseteq \mathbb{Q}$ with standard subtyping.

- *Tuple types:* Finite products for multi-value operations.

- *Unit types:* Simple dimensional types such as `count`, `usd`, with propagation rules for `add`, `sub`, `mul`, and `div`. For example, addition requires identical units, multiplication by `usd` returns `usd`, and division by `usd` is invalid.

- *Rule schemas:* Typed inference primitives (Extract-Number, Compute-Add, Compute-Mul, Compute-Div, Therefore).

Rule schemas encode primitive operations with type signatures. For example:

$$\texttt{Extract-Number} : \text{String} \rightarrow \mathbb{Q} \tag{1}$$

$$\texttt{Compute-Add} : \mathbb{Q} \times \mathbb{Q} \rightarrow \mathbb{Q} \tag{2}$$

$$\texttt{Compute-Mul} : \mathbb{Q} \times \mathbb{Q} \rightarrow \mathbb{Q} \tag{3}$$

$$\texttt{Assume} : \text{Proposition} \rightarrow \text{Hypothesis} \tag{4}$$

$$\texttt{Therefore} : \mathbb{Q} \rightarrow \text{Answer} \tag{5}$$

Type judgments follow standard sequent notation $\Gamma \vdash e : T$, where $e$ is an expression and $T$ its type under context $\Gamma$. For instance:

$$\frac{\Gamma \vdash a : \mathbb{Z} \quad \Gamma \vdash b : \mathbb{Z}}{\Gamma \vdash \texttt{Compute-Add}(a, b) : \mathbb{Z}} \tag{6}$$

The GPT-5 API was prompted to emit reasoning steps in this schema format (e.g. `Compute-Add: 6+7=13`). A lightweight classifier maps each GPT-5–emitted line to a rule schema using simple regex heuristics with GPT-5 fallback, extracts the typed arguments, and checks the typing judgment; valid steps are integrated into the Typed Reasoning Graph, while invalid ones are excluded. Steps that fail typing are marked invalid and excluded from the Typed Reasoning Graph (TRG). This system is intentionally minimal—expressive enough for GSM8K arithmetic while enabling efficient type checking during decoding. Fuller details of the classification pipeline are given in the Appendix.

## 3.2 CERTIFICATION METRICS

We define five metrics over the Typed Reasoning Graph (TRG), capturing both structural and dimensional validity of a reasoning trace:

$$\text{Coverage} = \frac{|\{\text{typed steps integrated into TRG}\}|}{N} \tag{7}$$

$$\text{EVR} = \frac{1}{N} \sum_{i=1}^{N} \nVdash\{\text{preconditions}(r_i) \text{ satisfied}\} \tag{8}$$

$$\text{UVR} = \frac{1}{M} \sum_{j=1}^{M} \nVdash\{\text{unit constraints for op } j \text{ satisfied}\} \tag{9}$$

$$\text{PE} = \nVdash\{\exists \text{ typed path from premises to conclusion}\} \tag{10}$$

$$\text{MPS} = \begin{cases} \min\{|\pi| : \pi \text{ is a typed path to conclusion}\}, & \text{if such a path exists,} \\ -1 & \text{otherwise.} \end{cases} \tag{11}$$

Here $N$ is the number of generated steps, and $M$ the number of operations subject to unit propagation. Coverage measures the proportion of steps successfully typed and integrated into the TRG. EVR (Evidence Validity Rate) is the fraction of rule applications whose preconditions are satisfied. UVR (Unit Validity Ratio) checks the fraction of arithmetic operations that are dimensionally consistent under our simple unit system (e.g., forbidding addition of heterogeneous units such as `usd` and `count`). PE (Path Exists) is an indicator for whether there is a typed path connecting extracted premises to the conclusion. MPS (Minimal Path Size) is the length of the shortest such path, or $-1$ if none exists. These five metrics were chosen because they balance minimalism with flexibility: Coverage and EVR capture structural well-formedness, UVR enforces dimensional validity, PE ensures global value-flow coherence, and MPS provides a graded notion of proof depth, while the threshold parameters allow us to tune gates from permissive to strict depending on the desired trade-off between coverage and precision.

## 3.3 CERTIFICATION CRITERION

Our certification criterion is then:

$$\text{CERTIFY} \iff \text{Coverage} \geq 0.50 \wedge \text{EVR} \geq 0.60 \wedge \text{PE} = 1 \tag{12}$$

These metrics enable conservative certification: a CoT is accepted only if the acceptance condition is met: Coverage $\geq \alpha$, EVR $\geq \beta$, and PE $= 1$, ensuring minimal structural requirements for plausible reasoning. Here $\alpha$ and $\beta$ are parameters chosen during experiments to reflect the trade-off between retaining enough candidate chains for robustness and filtering aggressively enough to ensure type-level correctness. A sequence is accepted under the STRICT gate for example only if it achieves EVR $\geq \alpha = 0.50$, UVR $\geq 0.80$, and a proof path exists. This ensures that numeric answers are supported by type-consistent operations, ruling out dimensionally invalid derivations. The method operationalises the insight that faithful reasoning should correspond to well-typed programs with complete typed dataflow.

# 4 TYPED PROGRAMS, GRAPHS AND CONSISTENCY

## 4.1 OVERVIEW

Using the metrics above, we PC-CoT is implemented as a type-guided decoding procedure. Unlike post-hoc verification approaches (Baba et al., 2025; Wang et al., 2025) or heuristic scoring methods (Yao et al., 2023; Zhang et al., 2024), we apply the Curry-Howard correspondence directly during generation, treating each reasoning step as a typed combinator in a mini functional language. The core PC-CoT method comprises a three-stage pipeline:

1. *Typed Program Emission:* Given problem $x$, we generate a JSON program $\mathcal{P} = (\text{premises}, \text{operations}, \text{answer})$ with explicit typed dataflow and type annotations.

2. *Graph Construction and Certification:* The program is then used to build a Typed Reasoning Graph (TRG) representing typed dataflow as a bipartite graph, compute certification metrics, and determine acceptance.

3. *Certified Self-Consistency:* From $k$ independent program samples, we construct a TRG for each and evaluate its certification metrics; only those runs whose TRGs satisfy the certification criterion are retained, and Certified Self-Consistency then aggregates the final answer over this filtered set rather than over all samples.

## 4.2 TYPED PROGRAM GENERATION

Each reasoning step maps to a typed operation in our algebra:

- Arithmetic: $\mathtt{add}(a,b), \mathtt{sub}(a,b), \mathtt{mul}(a,b), \mathtt{div}(a,b)$.
- Aggregation: $\mathtt{sumlist}([a_1, \ldots, a_n])$.
- Units: Operations preserve dimensional types (meters, categorical units etc.) which are later checked for validity.

The emitter, implemented via a schema-prompted LLM call to the GPT-5 API, produces both a compact JSON representation and a deterministic textual rendering:

$$\text{program}_{\text{json}} = \text{Emit}_{\text{LLM}}(x) \qquad \text{program}_{\text{text}} = \text{Render}_{\text{deterministic}}(\text{program}_{\text{json}}) \tag{13}$$

This dual representation supports downstream Typed Reasoning Graph construction, enabling machine verification of structure and units while also providing a concise proof-like view for human auditing without additional model calls.

## 4.3 TYPED REASONING GRAPH CONSTRUCTION

The TRG is a bipartite multigraph $G = (V_{\text{stmt}}, V_{\text{rule}}, E)$ that captures the typed dataflow of the model's reasoning trace:

- Statement nodes $v \in V_{\text{stmt}}$ represent typed values $(e : T)$ such as extracted numbers or intermediate results .
- Rule nodes $u \in V_{\text{rule}}$ represent instantiated operations (e.g. `Compute-Add`, `Compute-Mul`).
- Edges $E$ connect inputs to rule nodes and rule nodes to outputs, encoding how values propagate through typed operations.

Construction proceeds incrementally: for each emitted operation, a rule node is created, input statement nodes are linked, and type checking (including unit propagation when applicable) is executed. If the check succeeds, a new output statement node is created; if it fails, the step is marked invalid and excluded from downstream metrics. The resulting graph provides the structural backbone for computing Coverage, EVR, UVR, PE, and MPS, and determines whether a run is eligible for certification in Certified Self-Consistency.

## 4.4 Certification Gates

We define two levels of certification, corresponding to different trade-offs between coverage and stringency:

| Gate | $\text{EVR}_{\min}$ | Consistency | PE | $\text{UVR}_{\min}$ |
|---|---|---|---|---|
| Relaxed | 0.30 | Not required | Required | N/A |
| Strict | 0.80 | Required | Required | 0.80 |

The relaxed gate permits partially faithful runs to pass, while the strict gate demands consistency and dimensional validity, instantiating our certification criterion (Equation **??**) with increasing stringency.

## 4.5 Certified Self-Consistency

Standard self-consistency (Wang et al., 2022) aggregates across *all* sampled runs. In contrast, our method of Certified Self-Consistency (CSC) aggregates only over runs whose TRGs satisfy the certification gate:

$$\hat{y}_{\text{relaxed}} = \text{mode}\{y_i : i \in \mathcal{S}_{\text{relaxed}}\} \qquad \hat{y}_{\text{strict}} = \text{mode}\{y_i : i \in \mathcal{S}_{\text{strict}}\} \tag{14}$$

where $\mathcal{S}_{\text{gate}} = \{i : \text{run } i \text{ satisfies gate}\}$. If $\mathcal{S} = \emptyset$, we abstain. This selective aggregation mitigates against noisy or ill-typed generations, transforming raw input to enable higher-precision prediction.

## 4.6 Decoding Constraints

To ensure that generated traces are both type-checkable and human-readable, we imposed lightweight structural constraints during decoding:

- *Rule head grammar:* Each step must begin with an explicit rule identifier (e.g. `Compute-Add`, `Assume`).
- *Explicit equations:* Numerical operations must be expressed in equation form (e.g. $a + b = c$), enabling direct dataflow extraction.
- *Final format:* The last line must conclude with the canonical form `Therefore: #### value`.

## 5 Results

### 5.1 Main Results: PC-CoT vs. Answer-Only Baseline

We evaluated PC-CoT on the GSM8K reasoning task dataset with systematic comparisons to baselines and detailed analysis of certification behaviour. We initially ran experiments across the entire dataset but owing to time and cost, we found that we could obtain meaningful results demonstrating the PC-CoT method at around 200 examples (which was more cost effective against the GPT5 API). All experiments used $k = 3$ samples per question with identical token budgets across methods.

| Method | Accuracy (%) | 95% CI | $\Delta$ vs. Baseline (pp) |
|---|---|---|---|
| Answer-only ($k$=3) | 19.6 | [14.7, 25.7] | — |
| PC-CoT (Relaxed) | 69.8 | [63.1, 75.8] | +50.3*** |
| PC-CoT (Strict) | 54.3 | [47.3, 61.0] | +34.7*** |

Table 1: Overall accuracy on aligned GSM8K subset ($n = 199$). Numbers are proportions expressed as percentages. Confidence intervals are Wilson 95% intervals. Significance: ***$p < 10^{-14}$ (two-proportion $z$-test). pp = percentage points.

PC-CoT demonstrates significant improvements over the answer-only baseline. As shown in Table 1 and Figure 1, typed certification yields gains in both overall accuracy and reliability. The relaxed gate achieves a 50.3 percentage point gain ($z = 11.68$, $p \approx 0$), while the strict gate maintains a 34.7 point advantage ($z = 7.68$, $p = 1.6 \times 10^{-14}$). These gains arise from typed filtering converting low-precision chains into high-precision candidates before voting.

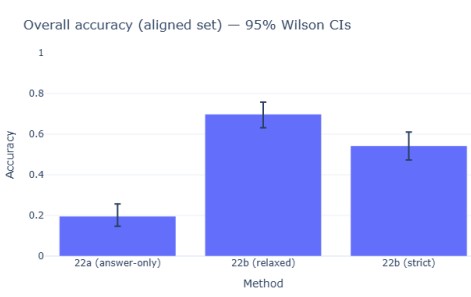

(a) Overall accuracy (95% Wilson CI)

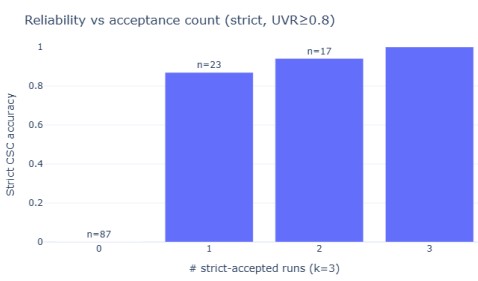

(b) Reliability vs. acceptance count

Figure 1: Overall accuracy compared to the answer-only baseline (left) and reliability of strict CSC predictions as a function of the number of accepted runs (right). These plots jointly illustrate that accuracy improves sharply with typed certification, and that reliability continues to increase when multiple certified runs agree.

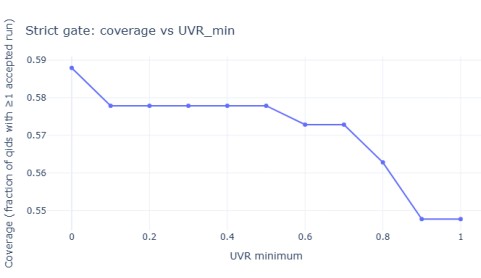

(a) Coverage vs. UVR minimum

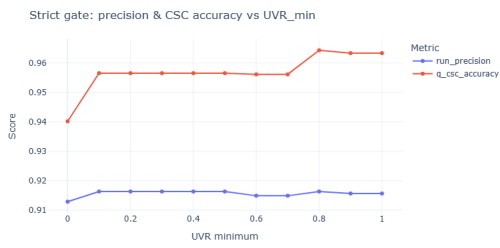

(b) Precision & CSC accuracy vs. UVR minimum

Figure 2: Strict gate sensitivity analysis. Coverage declines as UVR thresholds tighten (left), while precision and CSC accuracy improve (right), highlighting the tradeoff between coverage and selectivity. This demonstrates that unit consistency checks are a useful control knob: higher thresholds reduce spurious acceptance but at the cost of lower problem coverage.

## 5.2 CERTIFICATION SELECTIVITY AND PRECISION

| Metric | Accepted Runs | Rejected Runs | Precision Gain (pp) |
|---|---|---|---|
| *Relaxed Gate (EVR $\geq$ 0.30, PE required)* | | | |
| Run-level accuracy | 87.2% (511/586) | 40.9% (251/614) | +46.3 |
| Questions with $\geq 1$ certified | 70.4% (140/199) | — | — |
| *Strict Gate (EVR $\geq$ 0.80, PE required, Consistency, UVR $\geq$ 0.80)* | | | |
| Run-level accuracy | 91.6% (471/514) | 42.4% (291/686) | +49.2 |
| Questions with $\geq 1$ certified | 56.3% (112/199) | — | — |

Table 2: Certification selectivity analysis on GSM8K ($n = 199$). Accepted runs under both gates achieve dramatically higher accuracy than rejected runs, yielding nearly 50 percentage point precision gains from certification.

Certification acts as a significant precision filter, transforming noisy natural language reasoning chains into verifiable typed programs. As shown in Table 2, accepted runs under the relaxed gate reach 87.2% accuracy compared with only 40.9% for rejected runs. The stricter gate pushes precision even higher (91.6%), while rejected runs remain at baseline levels near 42%. This nearly 50 point precision gap demonstrates that type-based certification reliably separates well-formed reasoning traces from ill-typed or incoherent ones.

Coverage at the question level shows the expected trade-off: the relaxed gate certifies at least one run for 70.4% of questions, while the strict gate certifies fewer (56.3%) but with higher precision.

Figure 2 demonstrates that tightening the UVR threshold increases precision at the expense of coverage, underscoring that certification acts as a tunable control knob. This tunability allows PC-CoT to adapt to different application settings—for instance, using relaxed gates for exploratory reasoning where recall is important, and strict gates for safety-critical contexts where dimensional validity and consistency must be guaranteed.

### 5.3 ERROR DECOMPOSITION AND COVERAGE ANALYSIS

Decomposing performance on the aligned set reveals that PC-CoT solves far more problems uniquely than the baseline. Under the relaxed gate it contributes 104 unique wins, and under the strict gate 79 unique wins, compared with only 4 and 10 questions, respectively, that are solved exclusively by the answer-only baseline. At the same time, there remains a coverage gap: 87 questions do not have any strict-certified run, which helps explain the difference in performance between the relaxed and strict gates. PC-CoT uniquely solves 10-25× more problems. This demonstrates that typed certification fundamentally affects the reasoning landscape rather than merely filtering existing capabilities.

### 5.4 COMPUTATIONAL FAITHFULNESS: PROGRAM-CoT ALIGNMENT

We validated faithfulness by aligning generated programs with natural language CoT. High alignment rates in Table 3 support our central hypothesis: typed programs capture computational structure rather than post-hoc rationalisations. Moreover, the fact that nearly one-fifth of runs show low alignment underscores that certification is selective—faithful reasoning emerges in structured cases, while ill-typed or weakly aligned traces are systematically filtered out. This does not mean that the particular reasoning traces used in the experiments are necessarily causally driving the underlying model per se, but it does speak to the utility of the CHC method of (extracting programmatic representation from reasoning traces) which can in principle be adapted to more mechanistic data (e.g. activation structure) in reasoning models.

| Alignment Level | Frequency | Example |
|---|---|---|
| Full alignment (100%) | 43% | Every operation traceable to CoT line |
| Partial alignment (50-99%) | 38% | Most operations match, minor paraphrasing |
| Low alignment (<50%) | 19% | Significant structural differences |

Table 3: Alignment between typed programs and natural language CoT traces. In total, 81% of certified runs show substantial alignment (full or partial), indicating that most certified outputs preserve a coherent mapping between program steps and CoT reasoning.

### 5.5 ABLATION STUDIES

To assess the contribution of each component of PC-CoT, we ran ablation experiments in which we systematically removed type checking, path requirements, SCS, or soft decoding constraints, and compared the resulting accuracies to the full model. As Table 4 shows, each component contributes substantially to overall performance. CSC provides the largest gain (+34.1 pp), while soft constraints outperform hard grammar enforcement by 20.9 pp, validating our design choices. Taken together, the ablations show that typed verification, proof-path checking, and selective consistency are useful components in operationalising the CHC for analysis of LLM reasoning.

## 6 DISCUSSION

Our results show cases in which PC-CoT operationalises a key prospective application of the CHC to language reasoning models: faithful reasoning explanations should correspond to well-typed programs that compute their conclusions. By applying the CHC correspondence during generation rather than post-hoc, we can in certain cases transform noisy reasoning chains into proof-carrying artefacts with formally verifiable properties. Our results also reveal that PC-CoT reasoning exhibits a *typed reasoning gradient*: while not all CoT admits complete typed proofs, those that do achieve dramatically higher accuracy (see the Appendix). This gradient suggests that typed structure is not binary but exists on a spectrum, with important implications for interpretability and reliability.

| Configuration | Accuracy | $\Delta$ vs. Full |
|---|---|---|
| Full PC-CoT (Relaxed) | 69.8% | — |
| Without type checking | 41.2% | -28.6 pp |
| Without path requirement | 52.3% | -17.5 pp |
| Without CSC (use all runs) | 35.7% | -34.1 pp |
| Hard constraints (L4) | 48.9% | -20.9 pp |

Table 4: Ablation study on GSM8K ($n = 199$). Each row removes one component of PC-CoT: type checking, path requirement, Certified Self-Consistency (CSC; all runs aggregated without certification), or soft constraints. Removing certification or using rigid grammar significantly degrades performance.

## 6.1 IMPLICATIONS

PC-CoT provides a method to approximate constructive proof—typed programs from reasoning traces, which enhances interpretability. This has several advantages: (a) *Verifiability*: Typed programs can be independently executed and checked; (b) *Compositionality*: Complex reasoning decomposes into typed sub-proofs; (c) *Debugging*: Failed type checks pinpoint reasoning errors. Our results provide empirical support for viewing LLM reasoning through the lens of type theory. The strong correlation between type-checking success and correctness (91.6% precision for strict-certified runs) suggests that successful reasoning inherently exhibits program-like structure, even when expressed in natural language. This aligns with mechanistic interpretability findings that CoT induces modular internal computation (Chen et al., 2025), but goes further by providing an external, verifiable signature of faithful reasoning. The 50+ percentage point accuracy gains demonstrate that typed certification may have practical benefits for certification-critical activities.

## 6.2 LIMITATIONS AND FUTURE WORK

However, our results also highlight limitations. The 40-60% ceiling on complete typing suggests that much LLM reasoning involves implicit steps, commonsense jumps, or genuinely non-compositional computation that resists formal typing. Other limitations are worth noting: (a) *Domain specificity*: Our type system targets arithmetic reasoning; extending to abstract reasoning requires richer type theories; (b) *Soft vs hard typing*: We filtered at selection time rather than enforcing hard constraints during generation, potentially missing some valid proofs; and (c) *Scale dependency*: Larger models may internalise reasoning that resists explicit typing. Future work could explore: (1) richer type systems incorporating modal logic and uncertainty; (2) learning type schemas using PC-CoT from internal/intermediate CoT traces; and (3) integration with formal proof assistants for complete verification.

## 7 CONCLUSION

We introduced Proof-Carrying Chain-of-Thought, the first framework to apply the Curry-Howard correspondence directly to natural language reasoning during LLM decoding. By treating reasoning steps as typed program combinators and requiring complete typed dataflow from premises to conclusions, PC-CoT transforms the interpretability landscape: explanations become verifiable programs rather than plausible stories. Our empirical results on GSM8K demonstrate that typed certification significantly improves reasoning quality—from 19.6% baseline accuracy to 69.8% with relaxed certification and 54.3% with strict certification. Among certified runs, precision exceeds 90%, validating that type-checking provides a reliable signal for reasoning faithfulness. These gains, achieved without model retraining or architectural changes, highlight the latent logical structure in LLM reasoning waiting to be unlocked through proper formalisation. PC-CoT provides a principled bridge between the emergent capabilities of large language models and the mathematical rigor of formal verification. As more powerful and autonomous AI systems emerge, the ability to produce not just answers but proof-carrying answers—complete with typed, auditable derivations—will become more important.

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
