# OpenReview forum: "Typed Chain-of-Thought: A Curry-Howard Framework for Verifying LLM Reasoning"
_ICLR.cc/2026/Conference — ICLR 2026 Conference Withdrawn Submission_

### Official Review · Reviewer_wRYE · 2025-10-29

**Soundness:** 4
**Presentation:** 4
**Contribution:** 3
**Rating:** 6
**Confidence:** 4

**Summary:**

The paper describes an approach to certifying the faithfulness of a given Chain of Thought used to perform a numerical calculation, by translating a given CoT into a computational flowgraph of type operators in a functional language. Metrics are defined that are used to determine if the analyzed CoT can be certified, based on the generated graph's coverage of the full extent of the CoT, the degree to which preconditions of rules in the flowgraph are satisfied, and whether or not there exists a path from premises in the CoT through to the answer. In this manner, the process of certification can be used as a precision filter for a reasoning model's output on this class of problems (GSM8K is the benchmark used in the paper), and higher thresholds for coverage and precondition satisfaction yield improved accuracy at the cost of coverage.

**Strengths:**

In this reviewer's opinion, the application of the Curry-Howard isomorphism to LLM trustworthiness is innovative. The concept of faithfulness certification is new to this reviewer, and is well motivated. The connection between faithful reasoning and well-typed programs provides a principled foundation for verification. This theoretical grounding is particularly valuable as the field grapples with measuring and ensuring the alignment between LLM reasoning and reliable and repeatable computation.

The formalization in the paper is mathematically precise and concrete. The authors provide clear definitions of certification metrics including Coverage, EVR, UVR, PE, and MPS, establishing explicit certification criteria. While the type system and flowgraph language is limited to arithmetic operations, it is well-defined and implementable. The paper also demonstrates thoughtful analysis of trade-offs, clearly showing how different certification thresholds can be tuned for different application requirements, from exploratory reasoning where recall matters, to safety-critical contexts requiring validity.

**Weaknesses:**

The most fundamental limitation is the restriction to arithmetic operations. While the authors acknowledge this in their discussion, the challenge of extending even to basic logical reasoning appears substantial. The paper suggests richer type theories would be required, but doesn't adequately explore whether such extensions would remain tractable or maintain the same certification benefits.

**Questions:**

How does the certification handle the inherent stochasticity and variability in LLM outputs? If the same problem is presented with slight rephrasing or the model generates alternative valid reasoning paths, does the certification remain stable?

Given the Curry-Howard correspondence's natural connection to logical systems, what specific challenges prevent extending this approach to propositional or first-order logic problems? Is the limitation primarily in extracting formal structure from natural language, or does it lie in the computational complexity of type checking for richer logical systems?

---

### Official Review · Reviewer_gtX8 · 2025-11-01

**Soundness:** 1
**Presentation:** 2
**Contribution:** 1
**Rating:** 0
**Confidence:** 5

**Summary:**

This paper introduces Proof-Carrying Chain-of-Thought (PC-CoT), a framework applying the Curry-Howard correspondence to verify LLM Chain-of-Thought (CoT) reasoning. It types natural language steps into a formal program, constructing a Typed Reasoning Graph (TRG) to "certify" traces. The core contribution, "Certified Self-Consistency" (CSC), votes only over these certified traces. On a small subset of GSM8K (n=199), the authors claim a +50.3% accuracy gain (69.8% vs 19.6%) over an "answer-only" baseline.

**Strengths:**

Verifying CoT faithfulness is an important open problem in LLM interpretability and safety. Applying the CHC to verify natural language CoT (not just formal proofs). Applying CHC to verify natural language CoT is novel. Ablation studies are conducted to evaluate how components specify performance.

**Weaknesses:**

1. The main result of 50.3% improvement is against a non-optimized baseline. It does not compare against standard self-consistency (Wang et al. 2023), or standard chain of thought (Wei et al. 2023), which are the simplest trivial baselines. The authors should fix their experiment setup to include these.
2. The evaluation on ~200 samples from GSM8K is insufficient for ICLR, and cannot support the paper's claims of generality. The authors should evaluate on other arithmetic and non-arithmetic benchmarks to demonstrate generality.
3. Please consider using an external SMT solver such as Z3 or CVC5, similar to other techniques in this space such as Proof of Thought (Ganguly et al, 2024; 2025), SAT-LM (Ye et al., 2023), Logic-LM (Pan et al., 2023) and Logic-LM++ (Kirtania et al., 2024). Without an external solver, LLM based reasoning is prone to mistakes.

**Questions:**

Please see above weaknesses.

---

### Official Review · Reviewer_dvRW · 2025-11-02

**Soundness:** 2
**Presentation:** 2
**Contribution:** 1
**Rating:** 2
**Confidence:** 4

**Summary:**

The paper proposes **Proof-Carrying Chain-of-Thought (PC-CoT)**: a Curry–Howard–inspired way to treat a model’s natural-language reasoning as a typed program. During decoding, each CoT step is mapped to a rule schema (e.g., *Compute-Add, Extract-Number*), type-checked (including lightweight unit checks), and assembled into a **Typed Reasoning Graph (TRG)**. The system reports certification metrics (Coverage, EVR, UVR, Path Existence, Minimal Path Size) and accepts a trace only if thresholds are met; final answers are aggregated only over certified traces via **Certified Self-Consistency (CSC)**. On a 199 example subset of GSM8K with k=3, they report 69.8% (relaxed) and 54.3% (strict) accuracy versus 19.6% for answer only; ablations claim all components matter. Conceptually, it argues that a *faithful* CoT should correspond to a well-typed program, echoing “proofs ≙ programs, propositions ≙ types”.

**Strengths:**

* **Framing** A clean, teachable bridge from CoT to type theory (Curry–Howard). The limited type system (numeric, tuples, simple units) is pragmatic for GSM8K; the TRG abstraction is intuitive.

* **Clarity of pipeline**: The main pipeline (typed program emission → TRG → CSC) and the selectivity-vs-coverage trade-off are presented clearly in text and plots.

* **Method quality**: The TRG plus Coverage, EVR, UVR, PE, MPS metrics and gating criteria form a tidy, verifiable notion of well typed reasoning. CSC is a natural twist on Self Consistency that uses certification as a filter.

**Weaknesses:**

*	**Narrow evaluation and weak baseline**: Reporting only on an aligned GSM8K subset (n=199) and comparing against answer only is not sufficient. Needs CoT+SC, ToT/GoT, and tool-augmented math solvers on standard splits.

*	**Selection bias via Certification**: Voting only among “certified” runs boosts precision by filtering. Compare at equal token budgets and matched acceptance rates vs verifier-guided SC.

*	**Gate definitions (missing details)**: The strict gate includes a “Consistency” requirement, but the term is not clearly defined in the main text; also a dangling “Equation ??” appears where the certification criterion should be referenced.

*	**Reproducibility and cost**: The paper notes identical token budgets and k=3 sampling but does not quantify runtime/latency overhead for typing, TRG construction, and certification. The repository is mentioned, but end to end reproducibility (prompts, thresholds, seeds) is not guaranteed.

*	**Under-specified “aligned subset”**: The paper doesn’t define how the 199 GSM8K items were selected or whether they favor arithmetic with explicit units (benefiting UVR).

*	**Limited scope**: The type system and UVR checks are tailored to arithmetic with units; it’s unclear how well the approach carries to non unit or symbolic reasoning (e.g., *algebraic manipulation, logic puzzles*) or to tasks where typed quantities are sparse.

**Questions:**

1.	**Subset selection**: What exactly is the *“aligned GSM8K subset (n=199)”?* Please justify why this subset is representative. How do results change on the full GSM8K test set?
2.	**Baseline coverage**: Can you add **CoT + Self Consistency** and **ToT/GoT** with the same *k* and token budgets? If not, can you at least include a *vanilla* CoT+SC baseline to separate the effect of *certified voting* from general self consistency gains?
3.	**“Consistency” predicate**: Precisely define the “Consistency” check in the strict gate. Is it numeric agreement across steps, agreement between Therefore and computed outputs, or something else?
4.	**Sensitivity and ablations**: You show UVR threshold sensitivity. Could you also include sensitivity to EVR thresholds and to **k** (e.g., k=1,3,10)? Does CSC saturate with more certified runs?
5.	**Generalization**: Outside arithmetic (GSM8K), can your limited type system handle symbolic or commonsense reasoning? Any preliminary results on datasets where units aren’t salient? The limitations section hints at domain specificity; more evidence would help.
6.	**Failure modes**: Could you share qualitative cases where the method *rejects* correct reasoning due to typing gaps (e.g., *benign paraphrase, implicit steps*) and how you plan to reduce false negatives without relaxing gates too much?

---

### Official Review · Reviewer_szPE · 2025-11-02

**Soundness:** 1
**Presentation:** 1
**Contribution:** 1
**Rating:** 0
**Confidence:** 4

**Summary:**

This paper proposes Proof-Carrying Chain-of-Thought, a framework that applies the Curry-Howard correspondence to LLM reasoning traces. By treating reasoning traces as typed programs, it can improve LLMs' task performance on GSM8k and help with CoT interpretability.

**Strengths:**

- Working towards verified, faithful reasoning is definitely of interest to the LLM and AI reasoning community.
- In general, connecting ideas between programming language research and machine learning research can be productive.

**Weaknesses:**

- Sorry to be blunt, but this is a poorly written paper. There are many examples of unclear writing. For example, even the second sentence of the paper contains a significant conceptual error. LRMs are *not* characterized by CoT prompting. The whole point is that one trains LRMs so that they do not need to be prompted for reasoning (and their reasoning capabilities go far beyond prompting-based approaches).
- Similarly, many of the citations do not make sense. For example, why do the authors cite the Tree-of-Thoughts paper when they mention Graph-of-Thoughts?
- The introduction section links an anonymous repo, but the repo is empty.
- It seems that the authors evaluate GPT-5 on GSM8k, but the baseline only obtains 19.6% accuracy. This is clearly false, since much weaker models have achieved much better performance on GSM8k some time ago.

**Questions:**

In my opinion this paper is of poor quality. To what extent is it AI-generated?

**Details Of Ethics Concerns:**

This paper feels mostly LLM-generated.

---

### Note · Authors · 2025-11-12

I have read and agree with the venue's withdrawal policy on behalf of myself and my co-authors.